# Effect of Water Vapor on the Microstructure of $Al_2O_3$ on the Free-Standing MCrAlY Alloy at 1100 °C

**Minxuan Wu [1], Biju Zheng [1,*], Defeng Zhang [2], Taihong Huang [1], Chao Li [1], Qing Li [1], Wenlang Huang [1], Luyang Zhang [1] and Peng Song [1,*]**

[1] Faculty of Materials Science and Engineering, Kunming University of Science and Technology, Kunming 650093, China; wuminxuan@stu.kust.edu.cn (M.W.); taihonghuang@kust.edu.cn (T.H.); chaoli@stu.kust.edu.cn (C.L.); 20193130007@stu.kust.edu.cn (Q.L.); 20213130006@stu.kust.edu.cn (W.H.); zhangluyang@stu.kust.edu.cn (L.Z.)

[2] School of Automotive Engineering, Yancheng Institute of Technology, Yancheng 221051, China; dez@ycit.edu.cn

* Correspondence: 20110166@kust.edu.cn (B.Z.); songpeng@kust.edu.cn (P.S.)

**Abstract:** The oxidation resistance of the MCrAlY binding coat is due to the formation of protective $Al_2O_3$ oxide scale at high temperature. The oxidation behavior of NiCrAlYHf alloy in 1100 °C air and air-water vapor atmosphere was studied. The effect of water vapor on the microstructure and distribution of reactive elements was discussed. The results showed that the oxide scale in air has a double layer structure composed of columnar and equiaxed crystals, while the oxide scale in water vapor contains fine alumina grains, which provides more channels for the diffusion of reactive elements. In addition, The Cr element in the oxide scale is mainly concentrated in the outer equiaxed crystal zone, and the Hf oxide is mainly concentrated in the columnar crystal boundary. In air-water vapor atmosphere, the Cr element is uniformly distributed in the oxide scale.

**Keywords:** MCrAlY alloy; reactive elements; STEM; high temperature oxidation; alumina morphology





## 1. Introduction

MCrAlY (M = Ni, Co) was used as an alloy bond coat material with ceramic top coats to form a thermal barrier coating system. The MCrAlY coating alleviates the thermal mismatch stress between the substrate and ceramic top-coats. Moreover, MCrAlY coatings are commonly used as protective coatings for the high-temperature components of gas turbines and jet engines because they form an alumina scale with good compactness and adhesion at high temperatures [1–5]. The property and cohesiveness of the oxide scale is one of the main factors affecting the life of the coating [6–8]. Therefore, the free-standing MCrAlY was used to replace the alloy bonding coat in this study. The growth behavior of $Al_2O_3$ under high temperature oxidation was studied.

MCrAlY has been widely used due to its high temperature stability. In order to improve its high temperature oxidation resistance, many researchers have conducted relevant research. Previous studies have demonstrated that the oxidation resistance of a bond coat can be improved by adding appropriate reactive elements (such as Y, Zr, Hf, and La) [9,10]. Simultaneously, reactive elements have an important effect on the bonding properties of alumina scales. Researchers found that the addition of Y can improve the bonding properties of the oxide scale [11,12]. In addition, adding the reactive element Hf decreased the oxidation rate of CoNiCrAl alloys and enhanced bond performance [13]. Further study showed that the addition of Hf and Y had better effect on the performance improvement of MCrAlY coating. However, the role of Hf is greater than that of Y. Allam et al. showed that the oxidation weight gain of the alloy containing Hf was much lower than that of the alloy containing Y [14]. Studies have demonstrated that Hf is primarily concentrated on the alloy surface or grain boundaries. During oxidation, $HfO_2$ is formed and embedded in the

alumina scale. The formation of $HfO_2$ prevents external diffusion of Al, thereby reducing the growth rate of the alumina scale [15,16]. I. Milas et al. calculated the diffusion properties of reactive elements using density functional theory, and the results demonstrated that Hf and Y exhibit similar diffusion mechanisms in MCrAlY bond coats. The structure of oxide formed by Hf element is uniform and fine, while Y element tends to lead to a local thickening of oxide scale, which increases internal stress. In the process of high temperature oxidation, the diffusion path of Hf and Al is similar. As well, because Hf diffuses faster than aluminum, it will hinder the path of Al diffusion [17]. Due to the strong bond formation between Hf and the neighboring oxygen atoms in $Al_2O_3$, it can improve the adhesion between TGO and bond coat [18]. However, the oxidation characteristics of the reactive elements Hf and Y have not been fully investigated for various alloys. Furthermore, the relationship between the alumina microstructure and reactive elements needs to be studied.

The influence of humid air or water vapor-containing environments on the oxidation resistance and bonding properties of bond coats cannot be ignored [19]. Some researchers propose that the surface of the MCrAlY alloy is covered with a spinel structure in an atmosphere with water vapor. Water vapor is one of the important factors affecting the oxidation resistance of bond coats, and it can promote the formation of voids on the oxide scale/alloy interface to accelerate the spalling of the oxide scale [20]. In addition, water vapor affects the $\alpha$-$Al_2O_3$ phase transition at the beginning of oxidation (later on, oxide scales become compact and cohesive). [21,22]. However, other studies found that the oxidation rate of Nickel base alloys slows down in air containing water vapor. It forms a dense and uniform oxide scale on the surface of the alloy substrate, thus protecting it from environmental corrosion [23,24]. Other studies have demonstrated that the diffusion rate of oxygen at alumina grain boundaries is higher when polycrystalline alumina is exposed to water vapor at high temperatures [25,26].

Compared with other superalloys, nickel-based alloys have good thermal stability and low cost. Studies have shown that Al and Cr can stabilize the NiAl phase in the coating, and Al content in the range of 8 wt.% to 12 wt.% can play a good oxidation resistance [27]. The co-doping of reactive elements Y and Hf can improve the oxidation resistance and enhance the bond property. The doping content of Y and Hf is trace. Studies show that the content is 0.1~0.4 wt.%. Therefore, studying the oxidation characteristics of free-standing NiCrAlYHf alloys with an environment containing water vapor is of practical significance. To find a method for improving the oxidation resistance and bonding properties of MCrAlY alloy in air-water vapor, oxidation tests of NiCrAlYHf alloy in air and air–water vapor atmospheres are discussed in this paper. The effect of water vapor on the morphology and elemental diffusion distribution of alumina was studied along with its mechanism.

## 2. Experimental Procedures

NiCrAlYHf bar was smelted using a vacuum electric arc furnace. Its chemical composition is listed in Table 1. NiCrAlYHf samples with dimensions of 17 × 10 × 2 mm were machined from the alloy bar by spark erosion. These rectangular samples were then treated with sandpaper (P1200, SiC) to achieve the same surface roughness. Next, the samples were placed in an ultrasonic cleaning machine to remove impurities, such as oil from their surface. The side length and weight of the samples were measured using a Vernier caliper and a balance, respectively.

**Table 1.** The chemical composition of the free-standing NiCrAlYHf alloys (wt.%).

| Element | Ni | Cr | Al | Y | Hf | Ti | Fe | S |
|---|---|---|---|---|---|---|---|---|
| Content (wt.%) | 72.3 | 15.5 | 12 | 0.06 | 0.14 | 0.002 | <0.001 | <0.001 |

In order to study the influence of water vapor environment on the growth of alumina scales, two groups of samples were used in the experiment. In order to reduce experimental error, two samples of each group with the same conditions were measured at the same

condition. One group of samples was oxidized in the natural air of a high-temperature tubular furnace at 1100 °C, while the other group was oxidized in the air through water vapor. The water vapor atmosphere was provided by a generator outside the tubular furnace. Adjusting the temperature of water vapor to 60 °C, the amount of water vapor added corresponding to the saturation temperature of water vapor was about 25 wt.%. The as-received specimens were first heat-treated at 1120 °C for 2 h and then the oxidation experiment was carried out.

During the experiment, it was necessary to collect the oxidation weight change of the sample measured by the electronic balance. At the beginning of the experiment, the sample was directly placed into a furnace with a constant temperature of 1100 °C. After reaching a certain oxidation time, sample was removed from the high temperature furnace and we waited until it cooled to room temperature in natural air. the same sample mass was measured three times and the average of the results was calculated. The sample was taken out and measured every 3 h before oxidation for 30 h. Oxidation time at 30 to 100 h, it was measured every 10 h of oxidation. After 100 h of oxidation, it was removed and measured every 20 h.

After oxidation, all the specimens were examined by X-ray diffraction (XRD, Bruker D8) with Cu K$_\alpha$ radiation (wavelength = 0.15405 nm) in the 2θ range 10–90° with a step size of 0.02°. Surface and cross-sectional morphologies were observed by field emission scanning electron microscopy (SEM, FEI-Quanta 600, FEI Company, Hillsboro, OR, USA) with an energy dispersive X-ray spectrometer (EDS, Oxford INCAx-sight 6427, Oxford instrument, Oxford, UK). To protect the oxide scale during the sample preparation, a magnetron sputtering apparatus was used to sputter Pt at a current of 120 A under 4.5 Pa of pressure for 40 s. This was followed by Ni plating for 60 min at a current of 0.01 A, and the thickness was about 10 μm. The chemical composition of the oxide scale was analyzed using an electron probe microanalyzer (EPMA, JXA-8230, JEOL, Tokyo Akishima-shi, Japan). The grain characteristics and distribution of oxides were studied with transmission electron microscopy (TEM, Tecnai G2 F20 S-Twin, FEI Company, Hillsboro, OR, USA) and scanning transmission electron microscopy (STEM, Tecnai G2 F20 X-Twin, FEI Company, Hillsboro, OR, USA) coupled with EDS. Foil samples were prepared using a focused ion beam (FIB, Crossbeam 340, ZEISS, Oberkochen, Germany).

## 3. Results and Discussion

### 3.1. Oxides Morphologies and Phase Composition

The surface morphologies of the as-received specimens after surface polishing were observed by using SEM equipped with EDS (Figure 1).

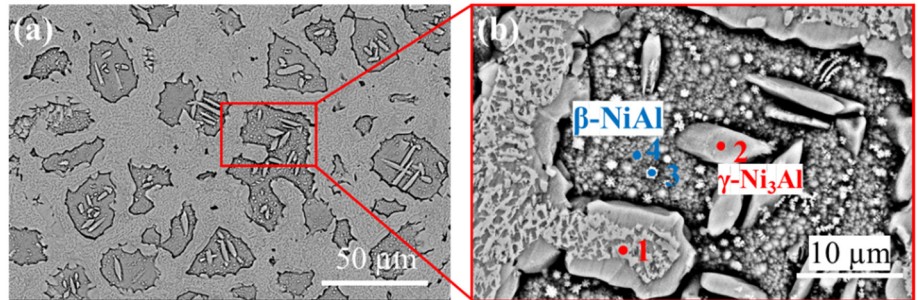

**Figure 1.** SEM/BSE microstructure of the as-received NiCrAlYHf alloy (**a**) and local enlarged image (**b**).

The chemical composition NiCrAlYHf samples are listed in Table 2. The as-received samples consisted of β-NiAl and γ-Ni₃Al phases. Figure 2 shows the surface morphology of the oxide scale on the NiCrAlYHf alloy after oxidation at 1100 °C for 100 h. The oxide scales were dense and almost uniform (Figure 2a,b). At a larger magnification (Figure 2c,e), needle-like alumina and block structure grains are present on the surface of the oxide scale

in an air atmosphere, and the oxide grain boundaries were clear for oxidized samples. In addition, a few bright white spots appeared. By contrast (Figure 2d,f), a large amount of white precipitate was present on the surface of the oxide scale oxidized for 100 h in the air–water vapor atmosphere. The abundant white deposits on the surface may be oxides of reactive elements. Moreover, their grain boundaries are so blurred that it is difficult to distinguish the morphology of individual grains.

**Table 2.** The chemical compositions (at.%) of the points in Figure 1 by EDS.

| Element | Ni | Cr | Al |
| --- | --- | --- | --- |
| 1 | 70.94 | 20.60 | 8.46 |
| 2 | 75.88 | 13.39 | 10.73 |
| 3 | 63.89 | 18.19 | 17.92 |
| 4 | 67.08 | 15.95 | 16.98 |

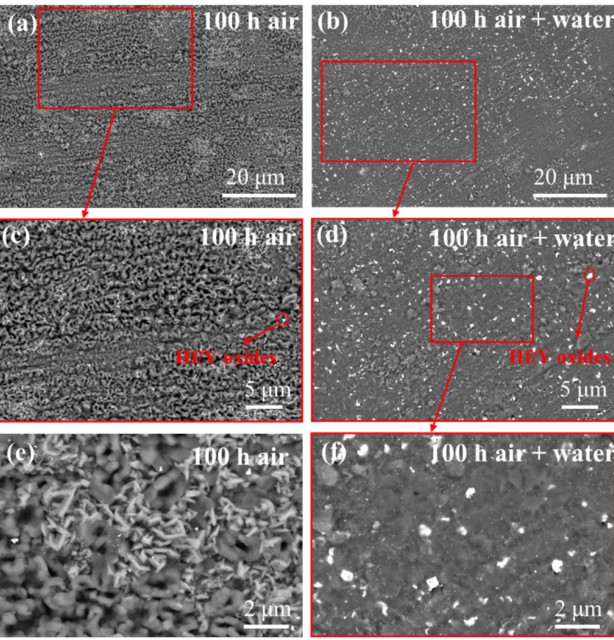

**Figure 2.** Surface SEM/BSE morphology of the oxide scale on NiCrAlYHf alloy after oxidation at 1100 °C for 100 h in air (**a,c,e**) and air–water vapor (**b,d,f**).

The cross-sectional SEM images of the samples oxidized for 100 h are shown in Figure 3. Dense and continuous oxide scales were formed on the alloy surface under different atmospheres [10,28]. However, the distribution of reactive elements during oxidation varied. Owing to the large partial pressure of oxygen on the outside, the reactive elements diffuse outwards and combine with oxygen to form oxides. Reactive element oxides existed in the oxide scale obtained in the air, while they were more dispersedly on the outer surface in the samples oxidized in the air–water vapor atmosphere. Thus, reactive elements can be observed outside the cross section of the oxide scale in an air-water vapor atmosphere, which is consistent with the observation of reactive elements on the surface in Figure 2b,d. Furthermore, the NiCrAlYHf alloy underwent internal oxidation under different oxidation conditions.

Figure 4 shows the XRD patterns of the samples oxidized for 12, 100, and 600 h. The results show that the oxide scale was mostly composed of alumina. Its composition in both atmospheres were similar. Hafnium oxides and spinels are present on the surface of the alumina scale. Notably, the transformation of NiAl to $Ni_3Al$ during the oxidation process is shown in the red box. This indicates that aluminum diffuses out of the substrate and combines with oxygen on the surface to form alumina.

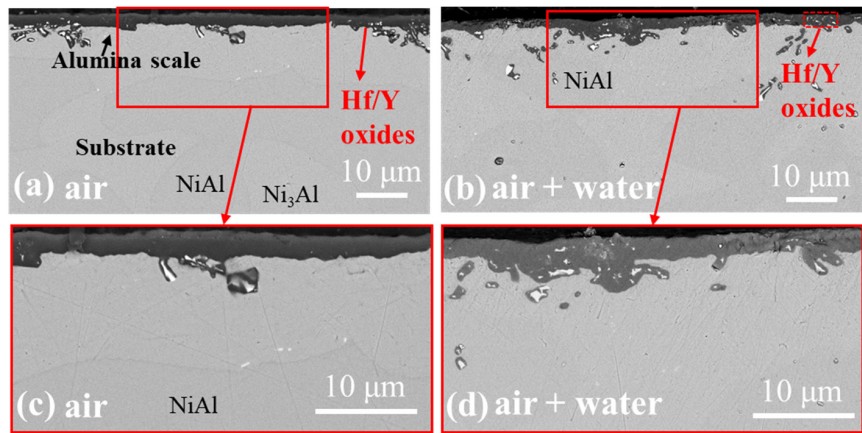

**Figure 3.** Cross-section SEM/BSE morphology of the oxide scale on NiCrAlYHf alloy after oxidation at 1100 °C for 100 h in air (**a**,**c**) and air–water vapor (**b**,**d**).

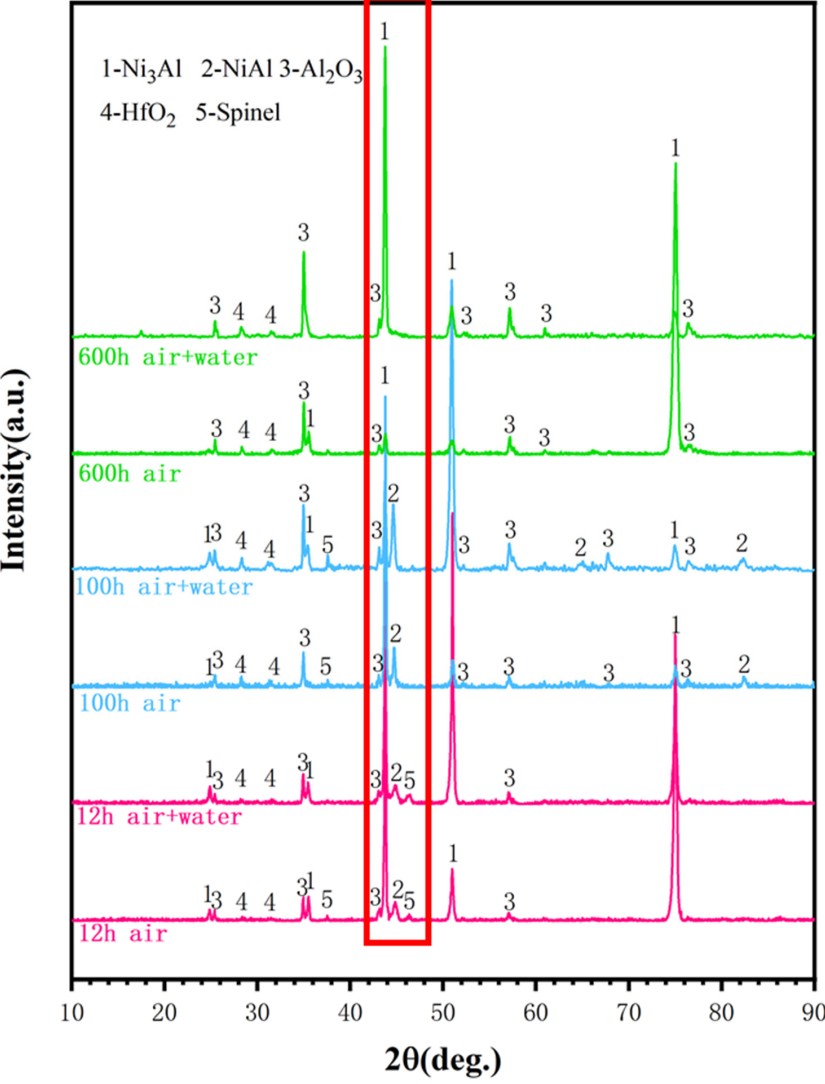

**Figure 4.** X-ray diffraction of the oxides on the NiCrAlYHf alloy at 1100 °C for 12, 100, and 600 h.

### 3.2. Oxidation Kinetics

The oxidation kinetics curve of NiCrAlYHf at 1100 °C in air and air–water vapor for 600 h is shown in Figure 5. The curve reflects the change in weight per unit surface area of the sample as a function of oxidation time. Overall, the weight gain trends in both

atmospheres were similar. According to Wagner's oxidation theory, the growth of oxide scale follows several laws, including linear law, parabolic law and logarithmic law [29–31]. Combined with the measured oxidation results, the oxidation kinetics of the alloy may follow the mixed rule. The oxidation reaction begins with a parabolic law and then changes to a linear law.

$$\Delta m = k_p \times t^n \tag{1}$$

where $\Delta m$ is the mass change in mg/cm$^2$, $t$ is the time in hours, and $k_p$ is the power-law rate constant [32,33].

As shown by the two vertical dotted lines in Figure 5, the oxidation process can be divided into the initial, middle, and late oxidation stages. The rate of the initial oxidation stage was the highest for both atmospheres. With the progress of oxidation reaction, the oxidation rate slows down in the middle stage of oxidation. This is because of the formation of alumina as a protective oxide prevents further reaction of oxygen, thus reducing the overall oxidation rate [34]. The later stage of stable oxidation was reached after approximately 450 h. At this point, the oxidation rates in the two atmospheres are almost the same. Both curves fluctuated, and the shedding of the oxide scale may have occurred.

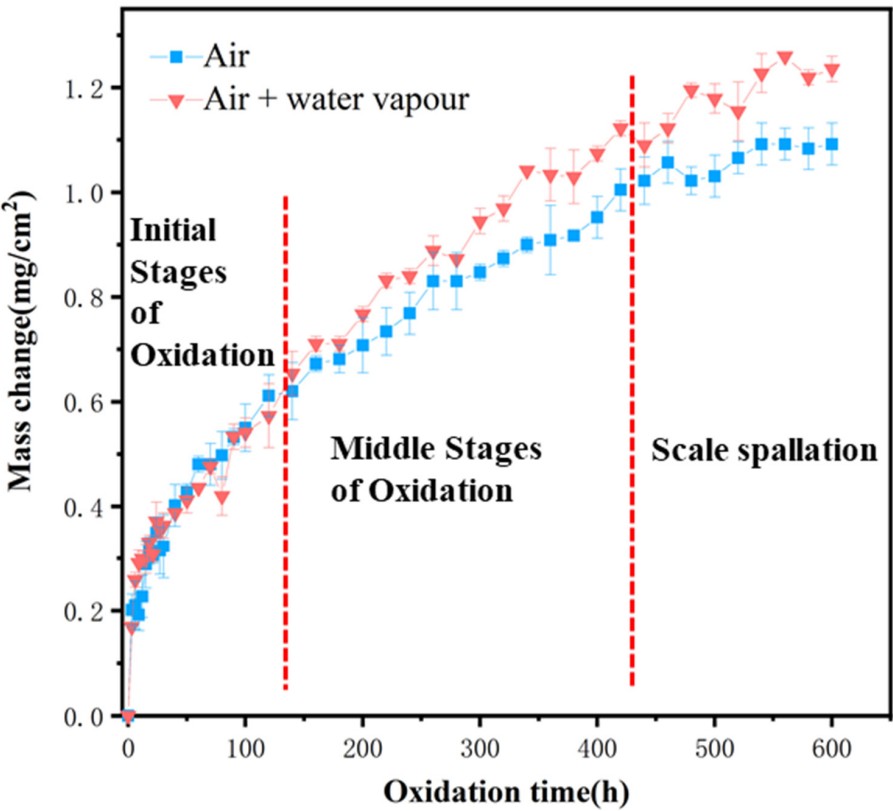

**Figure 5.** Mass gain curves of NiCrAlYHf sample after oxidation at 1100 °C for 600 h.

### 3.3. Analysis of the Composition of the Oxide Scales

Figures 6 and 7 show STEM images of the cross-section of the oxide scales in air and air–water vapor after 100 h. The difference in microstructures of the scales is apparent. The alumina scale had a prominent double-layer structure in the air. The inner layer is composed of neatly arranged columnar crystals, while the outer layer is composed of equiaxed crystals. This is consistent with the results of other's studies. On the contrary, no stratification is observed in the air–water vapor alumina scale (which is composed of the same alumina crystal and mixed reactive element oxides). The morphological differences of the alumina scales are discussed in Section 3.4.

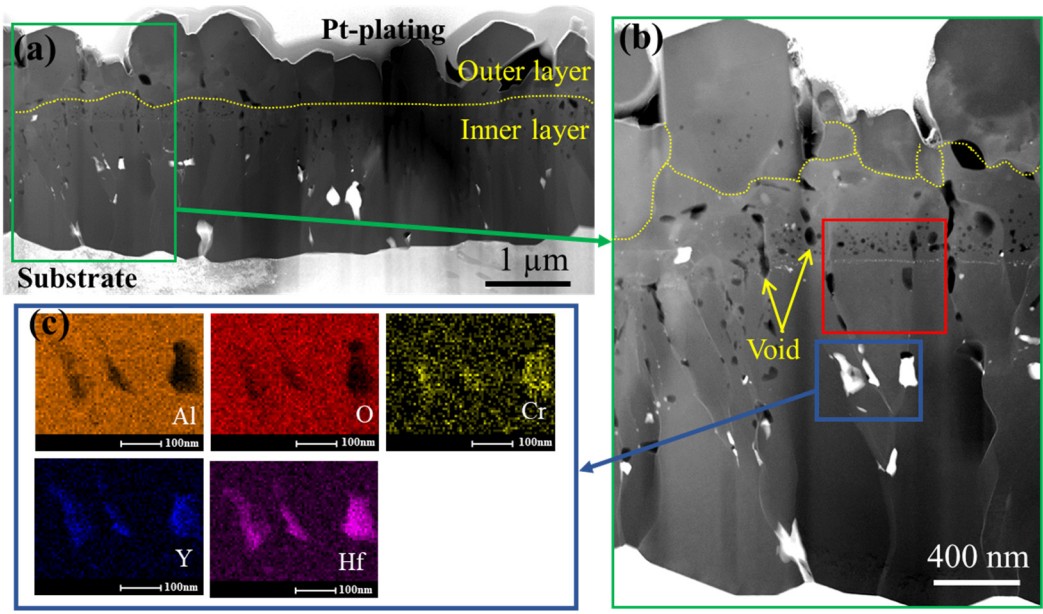

**Figure 6.** Morphology of the oxide scale on the NiCrAlYHf alloy (cut by FIB) after 100 h at 1100 °C in air (**a**), local enlarged image (**b**), and EDS analysis (**c**).

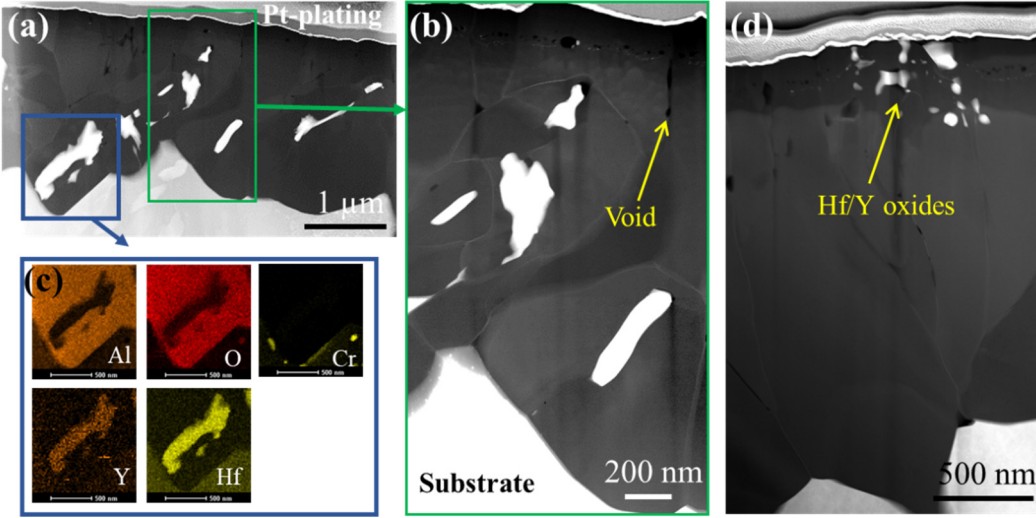

**Figure 7.** Morphology of the oxide scale on the NiCrAlYHf alloy (cut by FIB) after 100 h at 1100 °C in air–water vapor (**a**,**d**), local enlarged image (**b**), and EDS analysis (**c**).

Most of the white precipitates in the scales are distributed at the grain boundaries, with a small amount present in the lattice. As seen in the local magnification images (Figures 6b and 7b), the distribution of the white precipitates is different in the two atmospheres. The samples oxidized in the air show the white precipitates only at inner grain boundaries (Figure 6a). On the contrary, the white precipitate in the alumina scale obtained in the air–water vapor atmosphere is present throughout (Figure 7a). Moreover, as seen by EDS mapping, there are some reactive element precipitates in the oxide scale. The results are shown in the blue box in Figures 6c and 7c. The EDS results show that the precipitates on these grain boundaries are enriched with Y and Hf reactive elements. In addition, there are several voids at the grain boundaries of the alumina grains (Figures 6b and 7b). However, these voids are fewer in the air–water vapor environment. As discussed below, the differences in the distribution and size of the voids may be attributed to the effect of the alumina grains.

In this study, the results showed that there are several white precipitates on the surface of the oxide scale after air–water vapor oxidation for 100 h at 1100 °C. FIB was used to observe the microstructure morphology and distribution of white precipitates in the alumina scale under different atmospheres. As shown in Figures 6 and 7, there were significant differences. Figure 8 shows a local magnification of the microstructure morphology of the oxide scale. The EDS analysis results, combined with the markers in Figure 8, are shown in Table 3. Figure 8a shows that the oxide scales were mostly composed of alumina grains. The white precipitates were either oxides of $HfO_2$, $Y_2O_3$ or mixed oxide. In addition, $HfO_2$ and $Y_2O_3$ are enriched at grain boundaries. According to the diffusion theory of elements, Hf and Y diffuse outward by using the grain boundaries as diffusion channels.

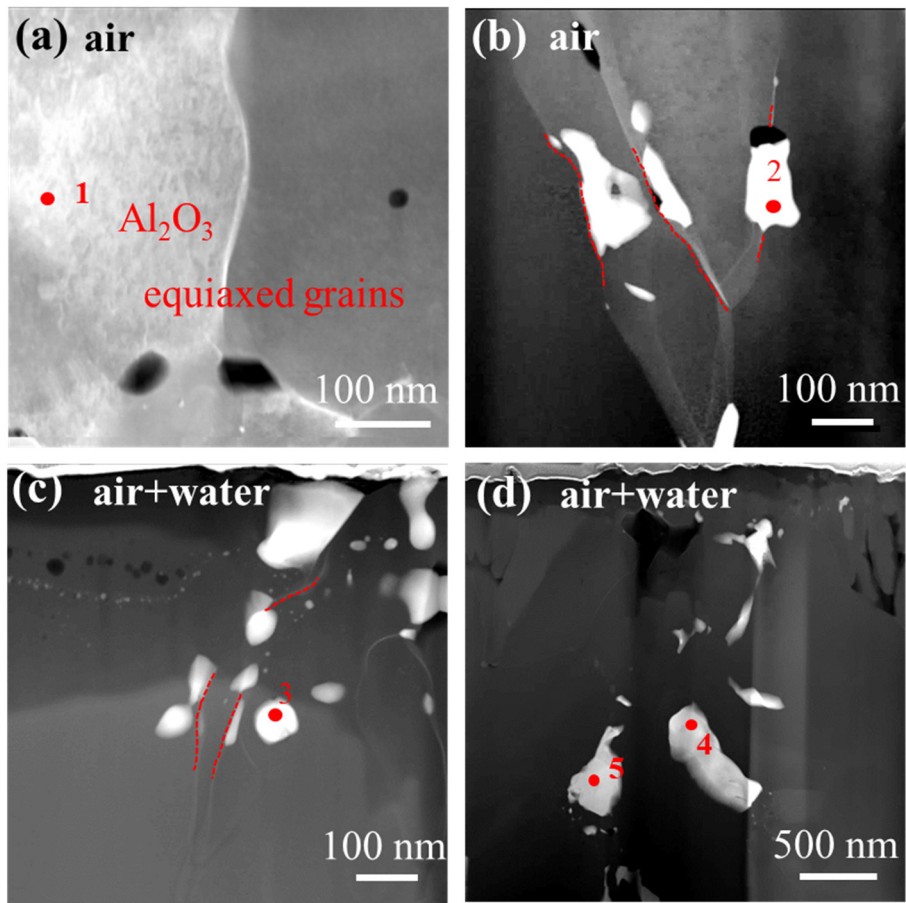

**Figure 8.** STEM microstructure of the NiCrAlYHf alloy (cut by FIB) after 100 h at 1100 °C: in air (**a**,**b**) and air-water vapor (**c**,**d**).

The oxides of the reactive elements (Hf, Y, and Cr) in the alumina scale diffused outward through the grain boundaries. This is confirmed by STEM images, which show bright white lines at the grain boundaries. This brightness originates from elements with an atomic number lager than aluminum. EDS was used to analyze the distribution characteristics of alumina grain boundary elements in the oxide scale exfoliated by FIB method. Figure 9a shows the grain boundary of the sample oxidized under air conditions. The grain boundary is near the substrate, which reflects the dynamic change trend of the element content along the line-scanning route. The aluminum and oxygen content at grain boundaries was lower than those at the crystal lattices on both sides. The concentration changes of Hf and Y shows their enrichment at grain boundaries. Notably, enrichment with Cr (4 at.%) was found at the outer grain boundaries (The red circle in Figure 9b). Figure 9c,d show the grain boundary of the sample oxidized in the air–water vapor atmosphere

(prepared the same way as above). The concentrations of Cr in the inner and outer grain boundaries were consistent and did not exceed 1 at.%. Notably, different distributions of Cr were observed in alumina scales under the two atmospheres. To avoid accidental selection of grain boundaries in line scanning mode, EDS mapping analysis was performed on the enlarged area of the inner and outer boundaries of the alumina scale (Figure 10). The alumina concentration in the entire oxide scale oxidized at 1100 °C in both atmospheres was consistent. In the air, the oxide scale is bounded by the outer equiaxed and the inner columnar crystals. The concentration of Cr in the outer layer is apparently higher than the one in the inner layer. This is consistent with the different Cr distributions.

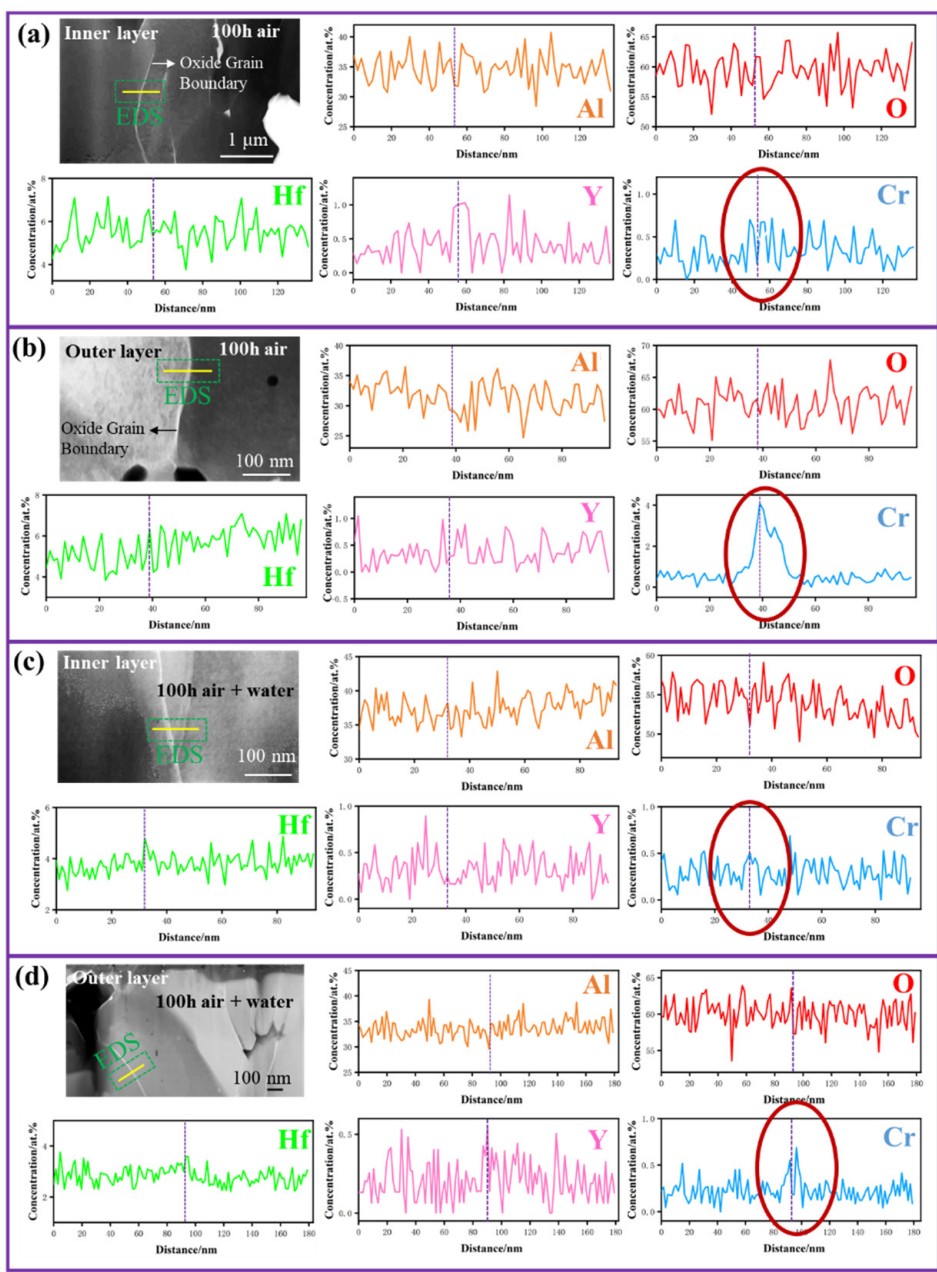

**Figure 9.** The morphology and EDS of the NiCrAlYHf alloy oxide scale after 100 h at 1100 °C: air, inner layer (**a**); air, outer layer (**b**); air–water vapor, inner layer (**c**); air–water vapor, outer layer (**d**).

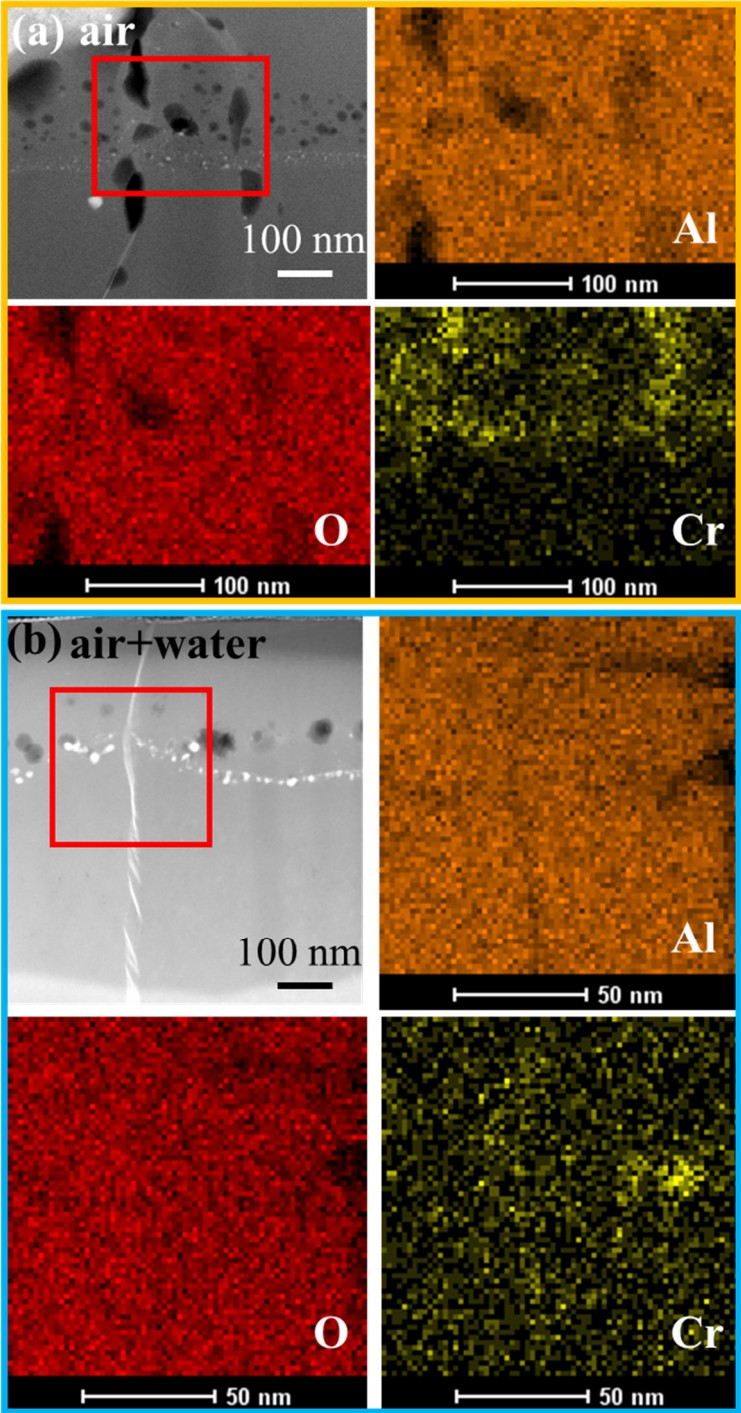

**Figure 10.** STEM microstructure and EDS-mapping results of the NiCrAlYHf alloy after 100 h at 1100 °C: in air (**a**) and air-water vapor (**b**).

**Table 3.** The chemical compositions (at.%) of the points in Figure 10 by EDS.

| Element | Al | O | Y | Hf | Cr |
|---|---|---|---|---|---|
| 1 | 41.92 | 57.82 | 0 | 0 | 0.25 |
| 2 | 0.73 | 61 | 2.48 | 35.77 | 0 |
| 3 | 0 | 58.11 | 9.32 | 32.55 | 0 |
| 4 | 0 | 65.49 | 0 | 34.50 | 0 |
| 5 | 0 | 63.04 | 0 | 36.95 | 0 |

EPMA was used to analyze the influence of water vapor from the macroscopic aspect of elemental diffusion. Figure 11 shows the results for the transition region from the outer alumina to the substrate. Figure 11a,b show the diffusion of elements in the oxidation process in air and air–water vapor, respectively. When internal oxidation occurs at the interface between the alumina scale and the substrate in an air atmosphere, the concentration of Cr fluctuated. However, this did not happen in the presence of water vapor. Therefore, it can be concluded that water vapor affects the diffusion rate of Cr during the high-temperature oxidation process.

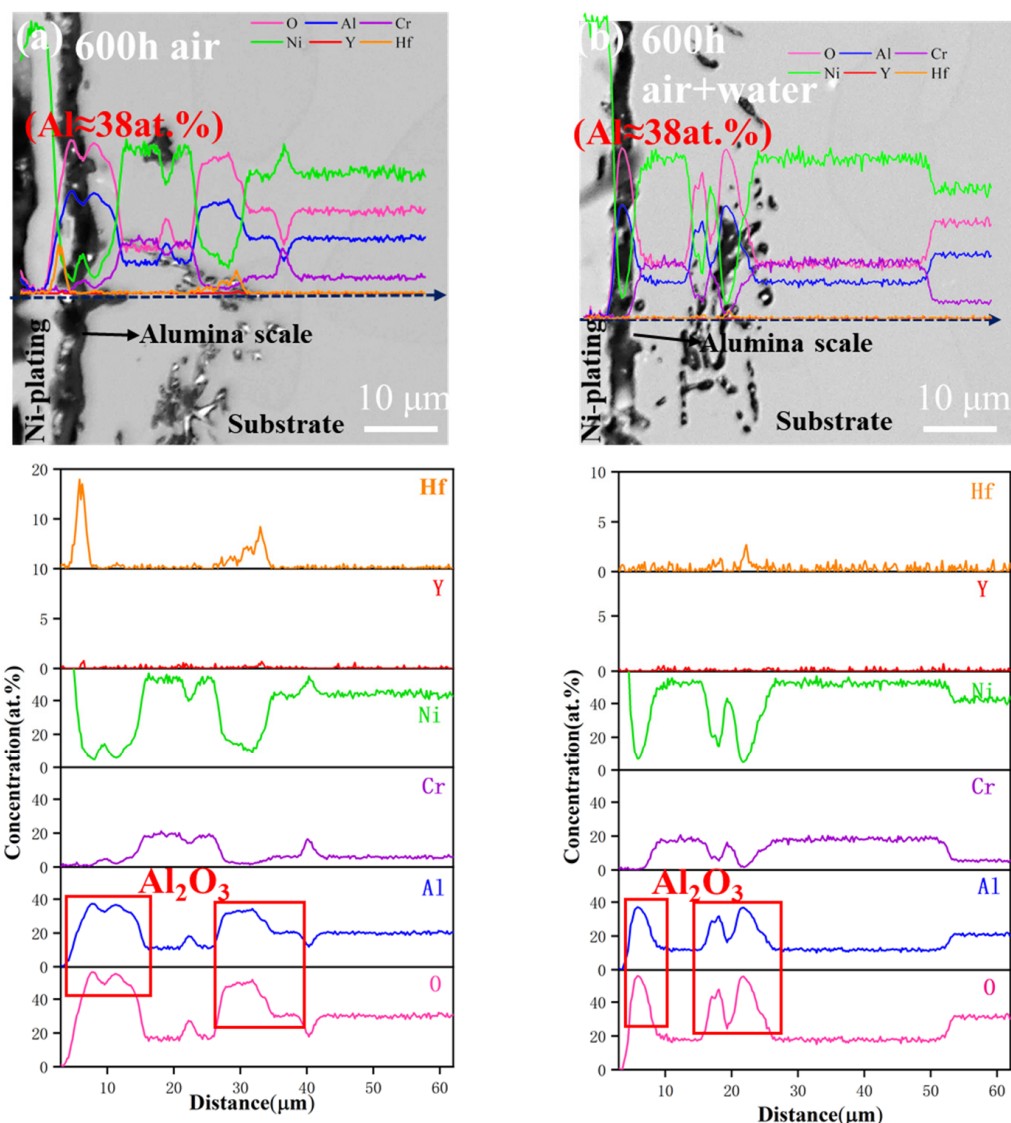

**Figure 11.** Morphology and EDS results of the NiCrAlYHf alloy oxide scale after 600 h at 1100 °C in air (**a**) and air–water vapor (**b**).

### 3.4. Microstructure Analysis of Oxide Scale

The STEM results (Figures 6 and 7) show the microstructure morphology of alumina scales in both atmospheres. In addition to finding differences in the distribution of reactive elements, the outer grain size of the alumina scale in the air is different. Some grain sizes are about 800 nm, while others are only 200 nm. Therefore, the fluctuation of the outer surface roughness was larger (Figure 6a). The outer surface of the scale in air–water vapor was relatively flat. The alumina scale on the left is thicker than that on the right. Because of the segregation of the reactive elements, a short-circuit diffusion path is formed, so the

alumina scale grows faster (Figure 7a). In conclusion, the outer grain size of alumina scale in air is larger than that in air–water vapor.

Figure 12 shows the atomic diffraction spots of the inner and outer grains in the two oxidation environments. The results in Figure 12a,b,e,f show that the alumina scale in the air is composed of hexagonal and orthogonal structures. However, the alumina scale in the air–water vapor environment comprises hexagonal grains. By comparing the difference in grain morphology of alumina in Figures 6 and 7, it was found that the outer grain size in the air is non-uniform, ranging from ca. 200 to 800 nm. The outer grains in the water vapor atmosphere were even smaller (200 to 500 nm). Owing to the volume difference of the particles, a large number of voids are easily formed during the accumulation of particles forming the oxide scale. This also explains the increased number of voids in the alumina scale after high-temperature oxidation in the air atmosphere (Figure 6).

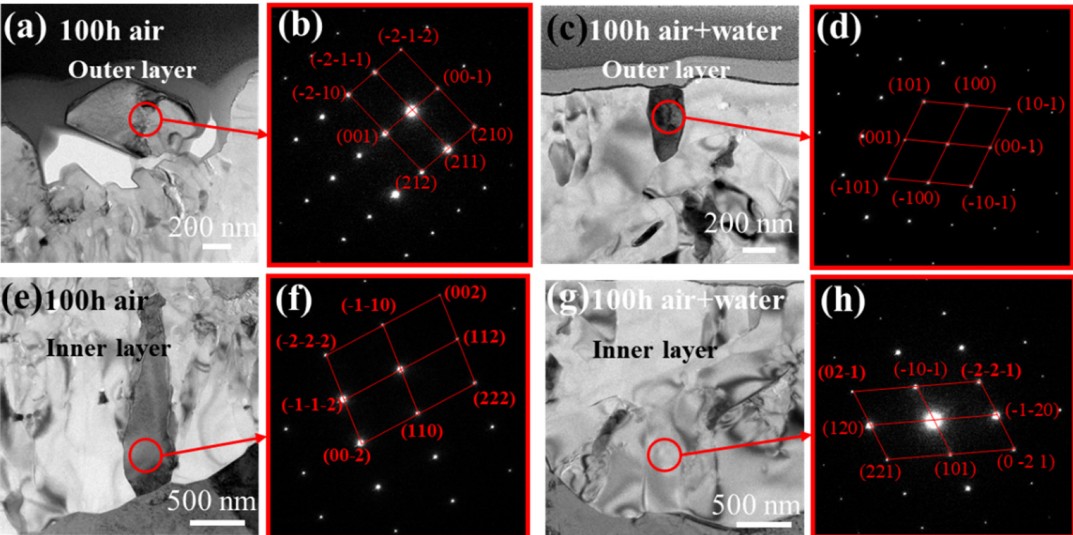

**Figure 12.** TEM image of alumina particle morphology and its diffraction spots after oxidation at 1100 °C for 100 h: air, in the outer layer (**a**,**b**); air–water vapor, in the outer layer (**c**,**d**); air, in the inner layer (e, f); air–water vapor, in the inner layer (**g**,**h**).

Next, the effect of water vapor on the crystal form of alumina is discussed. The TEM images clearly show that the alumina scale oxidized in air at 1100 °C is composed of grains of different sizes. However, the grains became uniform and fine when water vapor was added. Consequently, the cross-sectional surface of the alumina scale appeared smooth under the microscope. During oxidation in the air–water vapor atmosphere, Al in the substrate diffuses outward and reacts with $O_2$ to produce $Al_2O_3$. Simultaneously, Al reacts with $H_2O$ to form $Al(OH)_3$, which decomposes to $Al_2O_3$ at 1100 °C, as follows:

$$2Al + 6H_2O = 2Al(OH)_3 + 3H_2 \tag{2}$$

$$2Al(OH)_3 = Al_2O_3 + 3H_2O \tag{3}$$

During oxidation, the increase in nucleation due to the water vapor reaction leads to the slow growth of alumina grains, which fill the voids. Therefore, the scale structure was dense. However, the original grain grows constantly in the air atmosphere, while $Al_2O_3$ nucleation is decreased, resulting in an alumina scale with various grain sizes. There are more grain boundaries in the air–water vapor environment, which provides more channels for the diffusion of reactive elements. Moreover, these grain boundaries are conducive to the diffusion of Hf and Y, thus enriching more $HfO_2$ and $Y_2O_3$ on the outer surface of the alumina scale.

A schematic diagram of the influence of water vapor on the morphology and composition of the oxide scale is shown in Figure 13. At the initial oxidation stage, oxygen diffused inward, while aluminum in the substrate diffused outward [29]. Metastable $Al_2O_3$ and spinel phases were generated outside the oxide scale [35]. $HfO_2$ and $Y_2O_3$ were formed by a combination of reactive elements in the substrate because oxygen diffused inward along the grain boundaries between the alumina scale and substrate. As the oxidation progressed, the $Al_2O_3$ particles continued to grow, and the alumina scale thickness increased. The alumina scale in the air has a common double-layer structure: the inner layer is a columnar crystal, and the outer layer is an equiaxed crystal. The overall grain is non-uniform, and there is a large equiaxed alumina crystal on the outside. However, in the oxidizing air–water vapor environment, the nucleation of alumina grains is increased due to water vapor, so the surface of the alumina scale is composed of fine and uniform grains. Therefore, the crystalline form of alumina in water vapor is different from that in the air atmosphere. In addition, because of the impact of water vapor, the grains are small, and their boundaries increase. This provides more channels for the diffusion of reactive elements (i.e., Hf and Y) and enriches $HfO_2/Y_2O_3$ at the outer grain boundaries of the alumina scale. This enrichment limits the diffusion channels for aluminum and oxygen and can prevent contact between them. Consequently, there is a relatively slow oxidation rate in the air–water vapor environment in the initial oxidation stage. This is consistent with the oxidation kinetics results in Figure 5. In addition, the permeability of water vapor decreases the porosity of the alumina scale while influencing the alumina crystal shape. After a long period of oxidation, Hf and Y in the NiCrAlYHf alloy were consumed, and the oxide scale appeared spallated. Finally, the oxidation differences between the two environments gradually decreased.

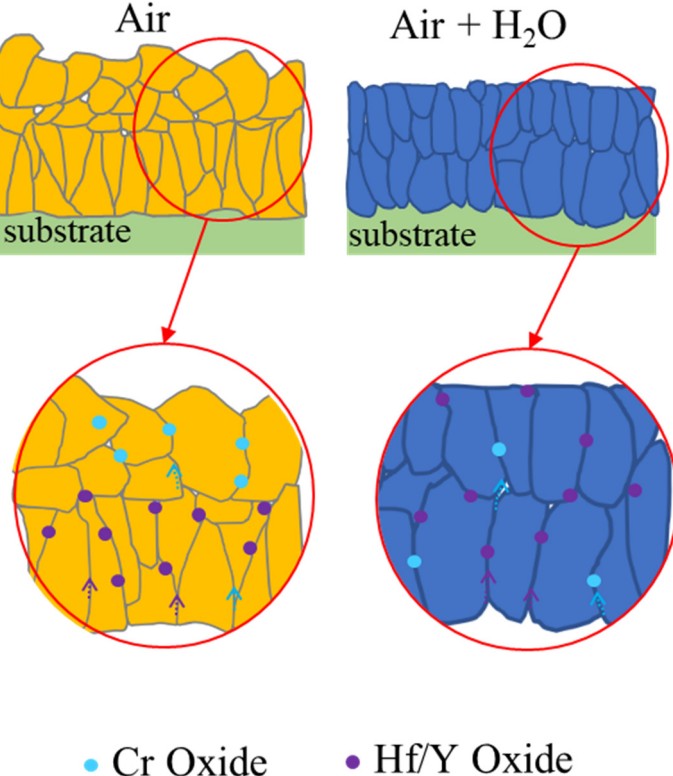

**Figure 13.** Schematic diagram of the morphology and composition of the oxide scale of NiCrAlYHf alloys at 1100 °C in different atmospheres.

## 4. Conclusions

The effect of water vapor on the morphology and reactive elements of the alumina scale on the free-standing NiCrAlYHf alloy after oxidation was analyzed. The following conclusions were drawn:

1. In the initial oxidation stage, the microstructure of alumina grains is affected by water vapor, which leads to an increase in the grain boundaries of the oxide scale. Consequently, it provides more channels for the diffusion of Hf and Y reactive elements. The mixed oxides of Hf and Y are only found on the inner side of the alumina scale generated in air. On the contrary, these mixed oxides can be found in the integrated alumina scale in an atmosphere containing water vapor.

2. In the oxidation process in the air–water vapor atmosphere at 1100 °C, Cr diffuses uniformly at the grain boundaries of the alumina scale. Water vapor affects the diffusion process of Cr. However, in the oxidation process in the air, most of Cr is enriched on the outer equiaxed grain boundary of the alumina scale, and a small amount is distributed on the inner columnar grain boundary.

3. The morphology of the alumina cross-section and the distribution of reactive elements depend on the oxidation atmosphere. In the air, the alumina scale has a two-layer structure consisting of outer equiaxed and inner columnar alumina grains. In the air–water vapor atmosphere, however, the alumina scale is composed of uniformly fine grains, and its stratification is not apparent.

**Author Contributions:** Conceptualization, M.W.; Data curation, P.S.; Formal analysis, W.H.; Funding acquisition, Q.L. and L.Z.; Investigation, D.Z.; Methodology, B.Z., T.H. and C.L. All authors have read and agreed to the published version of the manuscript.

**Funding:** This research was funded by the National Natural Science Foundation of China (No. 51961019), the Yunnan Province Science Technology Major Project (No. 2019ZE001).

**Institutional Review Board Statement:** Not applicable.

**Informed Consent Statement:** Not applicable.

**Data Availability Statement:** The data presented in this study are available on request from the corresponding author.

**Conflicts of Interest:** The authors declare no conflict of interest.

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
