# Peer review of "Effect of Water Vapor on the Microstructure of Al2O3 on the Free-Standing MCrAlY Alloy at 1100 °C"

_metals, doi:10.3390/met12050865_

Round 1
Reviewer 1 Report
The information from the first phrase in introduction “used as an alloy bond coat material with ceramic top coats to form a thermal barrier coating system” does not correlate with the information from the title “free standing”.
Line 36-37
Which is the difference between high-temperature oxidation rate and oxidation rate?
You must explain and cite why is the role of Hf grater than that of Y.
Line 39-41
The explained oxidation mechanism for Hf is similar with that of Y (segregation to grain boundaries). Why should Hf be better?
Please mention the company for the CoNiCrAlYHf bar.
You mentioned just gamma-Ni3Al, what about the gamma-Ni ?
Can you observe any difference concerning the beta(NiAl)-depleted zone in the CoNiCrAlYHf coating underneath the grown oxide scale?
Please explain changes of peak intensities of Ni3Al for the samples oxidized 100h in comparison with the other samples (see figure 4).
Please specify how many samples were investigated under the same conditions in order to generate the results presented in figure 5. I wonder if the difference between the samples tested under different conditions (air / air-water vapour) are within the standard deviation between the samples tested under the same conditions.
Why do you mention in conclusion “cyclic oxidation”?
I would expect some microstructural investigations of the samples exposed up to 600 h.
Reviewer 2 Report
Dear Authors,
Thank you for your submission. I find the topic interesting and of importance for different applications, mainly TBCs and similar. The research methodology and quality of data are both at good research level. However, the manuscript has also some important drawbacks which are mentioned below. So, I suggest the major revision in order to meet high-standards of the journal. My comments are as follows:
Abstract:
- Sentence: ‘Equipment such as XRD, SEM, FIB, TEM, and STEM are used for analysis.’ The equipment/research method is not that important itself. Please specify what was the studied and why.
- The last sentence of abstract is too general.
- English needs to be improved in the abstract.
Introduction:
- Lines 22-23, the first sentence should be revised, you did not spray top coat here and the TBC as a whole was not studied here.
- The sentence: ‘Moreover, it protects the substrate from corrosion by the external environment. [1-5]’ is not clear and should be revised. Citation is introduced wrongly, should be []. The reference to [1-5] is not clear, what is the point of including all these papers? The ref. 2 is about alumina coatings, this is just self-citation (!) and does not improve the scientific soundness of your manuscript.
- First paragraph of an introduction should be deeply revised, the scientific content is rather low here. Maybe it should be focused by discussing different ways of improving bond coat properties, including the idea of adding the reactive elements to MCrAl BC.
- Lines 31-32, the sentence should be revised. Why low cost and mature technology makes the MCrAlY of high research value? What is the point of ref. 10 and 11 here? Another self-citation.
- Lines 42-43, ‘Some researchers have calculated the diffusion characteristics of reactive elements’, please add references and more details.
- Lines 49-50, what do you mean by ‘mechanical strength of atomics’?
- English should be significantly improved in the Introduction part
Experimental procedures:
- How the chemical composition of NiCrAlYHf was evaluated here?
- Add information about equipment (inducing manufacturer), their accuracy, no. of samples studied here etc.
- How many samples were tested in booth oxidation tests?
- What is the point of heat treatment at 1080 C if the proper testing was performed at 1100 C?
- Add the thickness of Pt and Ni films.
Results:
- The labels marked with blue color in Fig. 1 and red color in Fig. 2 are not readable, please change.
- Lines 105-106, what is the point of BSE in the sentence?
- Lines 112-113, I suggest that the information about BSE and the contrast may be deleted.
- Lines 131-132, I do not understand the point on ‘many reactive elements presented in Figure 2’.
- Figure 3, alumina scale and Hf/Y oxides cannot be really distinguished from Figure 3. Please revise or add another micrographs with higher magnification.
- Figure 5:
- the SD bars are not presented. Considering that the mass gain is quite low, the trends should be supported by statistical data;
- why the mass change is given per cm2? These were free-standing coatings, maybe this should be presented as mm3 or cm3? - Line 167, specify ‘protective oxides’.
- Figure 7a, the green rectangle showing the zoomed area of Figure 7b should be labelled more precisely.
- Lines 194-195, the white precipitates are no really visible in Figure 7 but rather closer to the interface of substrate and oxides layer.
- Lines 210-211, revise the sentence
- Figure 9 caption, ‘air–water vapor, outer layer’ should be ‘d’ not ‘b’
- Lines 252-253, specify ‘different size’ of alumina scale size
- Line 258, what do you mean by ‘alumina flake’?
- Line 260, should be ‘a,b, e and f’.
- Line 280, specify the decomposition temperature
Editorial remarks:
- English has to be improved in the whole manuscript
- Citations must be improved and self-citations issues have to be handled
- References are introduced wrongly: ‘.[x]’ should be ‘[x].’
I suggest major revision but the above should be significantly improved.
Reviewer 3 Report
This work is focused on the analysis of high temperature oxidation phenomena in air or in water vapor-rich air of a MCrAlY alloy. The topic is very interesting, but the reported data are not well presented and dicussed.
The manuscript shows a poor English level: the language needs to be improved.
Several points have to be improved and modified:
The abstract is not well written and it is unclear. It should be rewritten.
- Introduction
- “Hf has higher activation energies, which hinder Al in the same diffusion path”; this sentence is not clear and it should be rewritten and well explained.
- “Because Hf forms strong bonds with oxygen atoms in the alumina scale, it improves the bonding strength between alumina and the bonding coating”; also this sentence is absolutely not clear, it should be rewritten and well explained.
- “In addition, Hf not only increases the bonding energy between interfaces but also increases the mechanical strength of atomics.”; this sentence is absolutely not clear and it should be rewritten and well explained.
- Experimental Procedures
- “the other group was oxidized in the air with 25 wt.% of water vapor (produced by a generator out-side the tube furnace)”. More details on test environment setup and control should be provided.
- Results and Discussion:
- “At a larger magnification (Figures 2c and 2d), needle-like alumina and block structure grains are present on the surface of the oxide scale in an air atmosphere”. The proposed SEM micrographs are not suitable to highlight “needle-like alumina and block structure grains”.
- Table 2 should be positioned just below the Fig.1
- “The needle-like alumina grains grow with time in the air, and a few bright white spots appear. These may be spinels of hafnium or yttrium oxides”. Top view EDS analysis on oxidized samples has to be performed.
- “the oxide grain boundaries were clear for samples oxidized in air, whereas they were blurred for samples oxidized in air–water vapor”. It is not so clear the presence of grain boundaries.
- “Reactive element oxides exist in the oxide scale obtained in the air, while they are more dispersedly on the outer surface in the samples oxidized in the air–water vapor atmosphere. Therefore, many reactive elements can be seen in Figure 2”. This sentence is not clear. The proposed SEM micrographs are not suitable to highlight this difference. Figure number is not 2, but 3. Moreover the box in Fig 3c indicating Hf/Y oxides is absolutely meaningless.
- “Notably, the transformation of NiAl to Ni3Al during the oxidation process is shown in the red box. This indicates that aluminum diffuses out of the substrate and combines with oxygen on the surface to form alumina”. The effect of NiAl loss due to TGO growth is not clear from the XRD results. The overlapping of peaks makes the understanding very difficult. Authors stated that “ aluminum diffuses out of the substrate and combines with oxygen on the surface to form alumina”: are you sure? Is this one the oxide growth mechanism?
- How did the authors collect the mass gain data? In the experimental sections the authors do not mention the cyclic oxidation procedure (cycle duration? Heating and cooling rate? Cooled down by using compressed air or natural convection? Please explain better.
- “The oxidation kinetics curve, as shown in Figure 5, combined with the classical oxidation theory, shows that the scale growth exhibits a parabolic dependence of scale thickness as a function of time” and “The oxidation kinetic data were fitted using Equation 2 to obtain a smooth oxidation curve”. The fitting results are not shown. The quality of the fitting is not presented (R2 value) and discussed. The fitting curves are not presented. So, there is no evidence that kinetics is parabolic.
- “The rate of the initial oxidation stage was the highest for both atmospheres. However, the rate of oxidative weight gain in the air–water vapor atmosphere was slower than the one in the air”. There is a substantial overlapping of the data sets and the authors do not provide any information about of the measurement errors (number of samples, number of collected data for each oxidation time, standard deviations) and the errors bars are not reported. How can the authors discuss about the comparison of oxidation rates?
- 6 a: It is not clear what is the inner and outer scale. Please introduce it and explain it in text.
- “the results show that there are several white precipitates on the surface of the oxide scale after air–water vapor oxidation for 100 h at 1100 °C.” … “Figure 8a shows that the oxide scales were mostly composed of alumina grains. The white precipitates were either oxides of HfO2, Y2O3 or mixed oxide”. The bright precipitates shown in Fig. 6 and 7 exhibit a lack of oxygen, according to eds maps. Now the authors state that these precipitates are oxides. It is very difficult to understand EDS results in table 3 if compared with EDS maps reported in fig 6 and 7. Moreover, what is the EDS resolution? Authors performed EDS point analysis on nano-precipitates. Is this analysis accurate?
- The paragraph 3.4 “Microstructure Analysis of Oxide Scale” can not be positioned at the end of the manuscript! The first part of description of TGO layer (rows 247-258) should be moved after Figg. 6 and 7.
Round 2
Reviewer 1 Report
The authors reviewed the manuscript as indicated and have justified some aspects which were not clear.
Reviewer 2 Report
Dear Authors,
In general, I am fine with the revision provided.
However, there are some editorial issues:
- Figure captions should be all checked and revised, for example:
- Fig. 1, a and b are not mentioned in the figure caption
- Fig. 2, there are six sub-figures (a-f) but only four mentioned in the figure captions (a-d)
- Fig. 5, it starts with (a) but there is only one chart in that figure, so no need for 'a'. Furthermore, do not start the figure caption with '(a)'
- Fig. 7, a-d in the figure and only a-c mentioned in the figure caption
- Fig. 8, Fig. 10 and Fig. 13 similar issues, please check carefully all figures and figure captions - Reference 13 is missing authors etc., please check carefully all references